Gut microbiota and its metabolites in Alzheimer’s disease: from pathogenesis to treatment

Zou Xinfu 1
Zou Guoqiang 2
Zou Xinyan 3
Wang Kangfeng 4
Chen Zetao 2022111281@sdutcm.edu.cn 1
1 Subject of Integrated Chinese and Western Medicine, Shandong University of Traditional Chinese Medicine , Jinan , Shandong , China
2 Subject of Traditional Chinese Medicine, Shandong University Of Traditional Chinese Medicine , Jinan , Shandong , China
3 College of Traditional Chinese Medicine, Hebei University , Baoding , Hebei , China
4 Traditional Chinese Medicine, Shandong University of Traditional Chinese Medicine , Jinan , Shandong , China
Tharmalingam Nagendran
Electronic publication date: 2024 Mar 13
Publication date: 2024
Volume: 12
Electronic Location ID: e17061
Received 2023 Dec 22; Accepted 2024 Feb 15
Copyright: ©2024 Zou et al.
Copyright year: 2024
Copyright holder: Zou et al.
License: This is an open access article distributed under the terms of the Creative Commons Attribution License, which permits unrestricted use, distribution, reproduction and adaptation in any medium and for any purpose provided that it is properly attributed. For attribution, the original author(s), title, publication source (PeerJ) and either DOI or URL of the article must be cited.
License URL: https://creativecommons.org/licenses/by/4.0/

Keywords: Alzheimer’s disease, Gut microbiota, Pathogenesis, Metabolites, Microbial-based therapeutics, Traditional Chinese medicine treatment

Funding: The National Natural Science Foundation Youth Foundation of China No. 81303028 Shandong Geriatrics Association Science and Technology Project No. LKJGG2021Y041 Shandong Geriatrics Association Science and Technology Project No. LKJGG2021W116 This work was supported by the National Natural Science Foundation Youth Foundation of China (No. 81303028), the Shandong Geriatrics Association Science and Technology Project (No. LKJGG2021Y041), and the Shandong Geriatrics Association Science and Technology Project (No. LKJGG2021W116). The funders had no role in study design, data collection and analysis, decision to publish, or preparation of the manuscript.

==============================
Introduction

An increasing number of studies have demonstrated that altered microbial diversity and function (such as metabolites), or ecological disorders, regulate bowel–brain axis involvement in the pathophysiologic processes in Alzheimer’s disease (AD). The dysregulation of microbes and their metabolites can be a double-edged sword in AD, presenting the possibility of microbiome-based treatment options. This review describes the link between ecological imbalances and AD, the interactions between AD treatment modalities and the microbiota, and the potential of interventions such as prebiotics, probiotics, synbiotics, fecal microbiota transplantation, and dietary interventions as complementary therapeutic strategies targeting AD pathogenesis and progression.

Survey methodology

Articles from PubMed and china.com on intestinal flora and AD were summarized to analyze the data and conclusions carefully to ensure the comprehensiveness, completeness, and accuracy of this review.

Conclusions

Regulating the gut flora ecological balance upregulates neurotrophic factor expression, regulates the microbiota-gut-brain (MGB) axis, and suppresses the inflammatory responses. Based on emerging research, this review explored novel directions for future AD research and clinical interventions, injecting new vitality into microbiota research development.

Introduction

Alzheimer’s disease (AD) is characterized by a gradual decline in cognitive function and loss of specific types of neurons and synapses. AD affects approximately 47.5 million people worldwide (GBD 2016 Neurology Collaborators, 2019). AD is characterized by progressive cognitive impairment, memory loss, recognition difficulties, and executive dysfunction and is more prevalent in individuals aged ≥65 years. AD is influenced by genetic and environmental factors. Given the global aging population, the number of AD patients is increasing by the year. It is estimated that there will be 82 million AD patients by 2030 and 152 million AD patients by 2050 (Meng et al., 2021).

The gut microbiota is important in human health and includes bacteria, viruses, and fungi. Recent studies demonstrated the influence of gut microbiota on the expression of host nervous system function (Jiang et al., 2017; Wu et al., 2017; Kumar et al., 2016; Sampson et al., 2016; Berer et al., 2017; Sharon et al., 2019; Solas et al., 2017). AD pathogenesis is complex and has not been fully elucidated. However, increasing evidence from biomedical research advances suggested the involvement of gut microbiota in host cognition and AD-related mechanisms (Kesika et al., 2021). AD is associated with gut microbiota dysbiosis, defined as altered microbial diversity resulting from disrupted microbial balance and the corresponding functional changes (Angelucci et al., 2019; Zheng et al., 2016). A recent systematic review indicated an association between antibiotic use and AD occurrence and progression, providing indirect evidence. This association is mainly due to the fact that broad-spectrum antibiotics influence gut microbiota composition and its biodiversity, and colonization is delayed after administration. Therefore, antibiotics might have an extensive and even contradictory role in AD (Angelucci et al., 2019). Additionally, the increasing AD prevalence in recent years was highly correlated with unhealthy diets and environmental exposures that affect the gut microbiota composition (Tan et al., 2019; Wojtunik-Kulesza, Oniszczuk & Waksmundzka-Hajnos, 2019).

Despite growing evidence of the role of gut microbiota in AD pathogenesis of AD, the exact role of dysbiosis remains unclear. Whether dysbiosis is an underlying causative factor or a result of AD-associated pathological changes remains unknown. Furthermore, gut microbiota research is shifting from association-based research to investigations of therapeutic interventions. Observational studies spanning several decades have highlighted the potential of gut microbiota modulation as a novel therapeutic strategy for AD. Ultimately, these studies aim to understand the underlying mechanisms and provide new insights into novel treatment strategies for AD (Chen et al., 2022). Furthermore, studying microbiota-gut-brain (MGB) axis involvement is essential for identifying new AD therapeutic targets and approaches (Megur et al., 2020). Therefore, it is crucial to focus on developing microbiota-targeted interventions while also considering the influence of the gut microbiota on drug treatment efficacy. This review presents a comprehensive overview of the link between dysbiosis and AD, the interaction between drugs and the gut microbiota during treatment, and the potential clinical translation of microbiota-targeted AD therapies.

Gut Microbiota Composition and AD

Gut bacterial microbiota dysbiosis and AD

Increasing evidence suggests a correlation between gut microbiota dysbiosis and AD. The gut microbiota composition of AD patients differs from that of healthy people, specifically regarding the microbial diversity and proportions of specific bacterial species (Hung et al., 2022; Wasén et al., 2022). A clinical study reported that AD patients had higher abundances of Deferribacteres, Bacteroidetes, and Phascolarctobacterium than the healthy population. However, the AD patients had lower abundances of Clostridiaceae (1.53% vs. 3.89%, P = 0.002), Lachnospiraceae (14.26% vs. 19.29%, P = 0.006), and Ruminococcaceae (8.08% vs. 14.25%, P = 0.019) (Liu et al., 2019). Various studies reported that the interaction between microbiota and the host immune system becomes imbalanced due to aging and disease. This imbalance increases proinflammatory bacteria and decreases anti-inflammatory bacteria (Pellegrini et al., 2023). This pattern is similar to opportunistic species in other neurologic disorders and various inflammatory bowel diseases (Cammann et al., 2023), supporting the hypothesis of immune inflammation in AD. Vogt et al. (2017) reported decreased microbial diversity in the gut microbiota of AD patients. Certain major bacterial phyla, such as Firmicutes and Clostridium, were the most affected. AD featured a decreasing trend in Actinobacteria and an increasing trend in Bacteroidetes, while Ruminococcus abundance was downregulated and Escherichia/Shigella abundance was upregulated (Cattaneo et al., 2017). Gut microbiota pathogenicity increases with life progression and the decline of various host functions, resulting in differential expression of microbial characteristics among different AD patients (de Cena et al., 2021; Martino et al., 2022) (Table 1). Furthermore, the microbial composition of AD patients exhibits regional heterogeneity. The highest proportion was found in a human gut bacterial sequence (ADAS) related to AD in Chinese and United States populations (Paley, 2019). However, there was no definitive evidence to establish a causative connection. Fecal microbiota transplantation (FMT) experiments in rodent models suggested a potential causal relationship between AD and gut microbiota balance and dysbiosis (Wang et al., 2021a; Nandwana & Debbarma, 2021). Gut microbiota variations trigger AD onset, while AD pathological changes disrupt the ecosystem by altering the gut environment. Therefore, AD and the gut environment interact closely and are mutually influential (Nguyen, Cho & Lee, 2023; Liu et al., 2020; Leblhuber et al., 2021; Simão et al., 2023).

Table 1 Gut microbiota composition in patients with Alzheimer’s disease.

Number	Gut microbiota composition in patients with Alzheimer’s disease	Reference	
1	Firmicutes, Actinobacteria phylum and Eubacteriumrectale↓; Bacteroides and Shigella ↑	Cattaneo et al. (2017)	
2	Prevotella, Sutterella, Haemophilus↓	Paley (2019)	
3	Escherichia-Shigella and Desulfovibrio↑	Chen et al. (2020), Borsom et al. (2023)	
4	Pseudomonas, Faecalibacterium, Fusicatenibacter, Blautia, and Dorea↑	Xi et al. (2021)	
5	Candidatus Saccharibacteria phylum and Proteobacteria ↑	Heravi, Naseri & Hu (2023)	
6	Saccharibacteria, Anaerotruncus, Vampirovibrio, and Alistipes  ↑;Dorea, Anaerostipes, Hallella, and Ruminococcus↓	Favero et al. (2022)	
7	Helicobacter pylori↑	Xie et al. (2023)	

The underlying pathological mechanisms of AD

Although overall changes in the gut microbiota of AD patients have been suggested, the specific pathogenic agents should be identified using further investigations. Experimental studies suggested that the aggregation of misfolded proteins, such as amyloid-beta and tau, and increased bacterial amyloid, such as curli fibers, cause neurodegenerative changes, significantly contributing to AD development (Chen et al., 2016; Pisa et al., 2015; Pluta et al., 2020). Experimental studies indicated that the human metabolites ‘canine uracil’ and myopeptide elevated DNJ-12 and DNJ-19J protein levels in an ‘AD rodent model’. These increased protein levels triggered the cytoplasmic unfolded protein response to eliminate A β42 aggregates and their associated toxicity. However, the ‘rodent model’ demonstrated that the genus Prevotella directly enhanced protein stability (Joshi et al., 2021; Walker et al., 2021). Furthermore, it was suggested that gut microbiota changes in AD mice are accompanied by corresponding changes to the V3 and V4 regions of the 16S ribosomal RNA genes (Shen, Liu & Ji, 2017).

Numerous human studies have demonstrated that bacteria, such as Helicobacter pylori, Borrelia burgdorferi, and Chlamydia pneumoniae, increase AD susceptibility by promoting tau protein hyperphosphorylation and elevating proinflammatory bacteria (Escherichia/Shigella) and decreasing anti-inflammatory gut microbiota (Ruminococcus) (Murray, Kemp & Nguyen, 2022; Cryan et al., 2019). Several experiments on AD mice reported increased abundance of Escherichia coli-Shigella and Desulfovibrio and an according increase in the abundance of Enterobacteriaceae, Pseudomonas, Clostridium, Prevotella, and Clostridium cluster IV RNA, contributing to amyloid-like protein deposition and microglial cell accumulation in the brain, triggering inflammatory responses and participating in AD pathogenesis (Chen et al., 2020; Borsom et al., 2023). Additionally, Xi et al. (2021) found an increased abundance of Pseudomonas and Faecalibacterium and a correlation between the higher abundance of the family Lachnospiraceae (Fusicatenibacter, Blautia, and Dorea) and more severe AD symptoms through non-targeted analysis of gut microbiota in AD patients’ fecal samples. Furthermore, a study using metabolic methods reported increased Candida and Proteobacteria in an AD mouse model. The results suggested a connection between gut microbiota imbalance and AD occurrence and progression (Favero et al., 2022).

Fujii et al. (2019) reported that transplanting fecal samples from AD patients into germ-free mice and subsequently administering this strain orally impaired the cognitive abilities of the mice. However, these specific pathogenic factors might interact with other contributing factors within the complex gut microbiota ecosystem rather than being limited to singular infectious elements. Therefore, examining the interplay and potential mechanisms of various microorganisms, including bacteria and fungi, and their respective roles in AD from different perspectives is crucial. The interactions between microbial populations might exhibit diverse complexities. Current research suggests that certain common bacteria and fungi influence key microbial communities within the body by forming symbiotic relationships or producing metabolites. A deeper understanding of the higher-order interactions that produce novel effectors in the microbial–host interaction might enable a more comprehensive understanding of the mechanisms underlying AD (Hu, Wang & Jin, 2016).

Gut non-bacterial microbiota dysbiosis and AD

AD pathogenesis extends beyond gut bacterial dysbiosis and involves non-bacterial microbiota. Bacteriophages contain randomly uniquely encoded peptides/proteins that can influence gut microbiota genotypic and phenotypic diversity, affecting microbial community functionality and contributing to AD development (Shkoporov, Turkington & Hill, 2022; Attai et al., 2022; Zhang et al., 2022b). Additionally, gut fungal ecosystem disruption was reported in AD patients, and the fungal–bacterial network function appeared to be associated with AD pathology (Castillo-Álvarez & Marzo-Sola, 2022). Previous studies have demonstrated the involvement of C17 fungal isolates (belonging to the genus Candida) in AD pathophysiology (Cruz-Miranda et al., 2020). The interactions between bacteria and non-bacterial microbiota in the gut are highly complex. Thus, studying only the bacterial characteristics without considering the role of other non-bacterial microbiota would distort the fungal–bacterial network function and limit research direction development, hindering understanding of the mechanisms underlying AD.

Gut microbiota-derived metabolites and AD

The gut microbiota–host interaction involves various important mediators crucial in the gut. These mediators include the direct products of multiple microbial groups (produced through bacterial fermentation of dietary fiber), bacteria-secreted compounds, and bacterial metabolites (D-glutamate and neurotransmitters) (Kesika et al., 2021; Li et al., 2019; Chang, Lin & Lane, 2020; Qian et al., 2021; Morrison & Preston, 2016; Wang, 2023). Gut microbiota compositional alterations in AD patients drive microbial metabolome changes, which are involved in AD pathogenesis. Bile acids (BAs) are closely associated with AD (Qian et al., 2021), where primary BAs (CA) are significantly decreased in AD patients, while secondary BAs and their conjugated forms (including DCA, GDCA, TDCA, and GLCA) are elevated (Mahmoudian Dehkordi et al., 2019; Alzheimer’s Association, 2019). BAs regulate the host immune response and directly affect the balance of T helper (Th)17 and T regulatory (Treg) cells (Hang et al., 2019). Tauroursodeoxycholic acid (TUDCA) rescued neuronal integrity loss by modulating γ-secretase activity and alleviated symptoms in AD patients (Nunes et al., 2012; Lo et al., 2013). The neurotransmitters DA, NE, His, serotonin, or GABA are directly or indirectly produced by gut microbiota, regulate brain function through blood circulation or neural transmission, and are fundamental in AD (Qian et al., 2021; Wang, 2023).

The relationship between AD and branched-chain amino acids (BCAAs) consisting of valine, leucine, and isoleucine has been confirmed (Qian et al., 2021). The BCAA concentration negatively correlated with the AD risk (Tynkkynen et al., 2018), and feeding AD mice with food containing BCAAs aggravated cognitive dysfunction (Li et al., 2018). Additionally, short-chain fatty acids (SCFAs), including acetate, propionate, butyrate, valerate, and hexanoate, participate in the AD pathophysiological processes by mediating immune inflammatory reactions through the modulation of prostaglandins, T cells, and TNF-α (Gill et al., 2018; van de Wouw et al., 2018; Doifode et al., 2021). Other microbial-derived metabolites, such as lipopolysaccharides (LPS) and amyloid protein fibers, are also involved in AD pathogenesis (Li et al., 2019; Chang, Lin & Lane, 2020; Qian et al., 2021). The levels of these metabolites somewhat reflect gut microbiota metabolic capacity. In many cases, the gut microbiota is a key factor influencing human health and disease. In addition to the gut microbiota composition, the functionalities of microbes and their products are also significant. Hence, disease research should comprehensively consider all aspects of microbes and their derivatives.

The MGB Axis in AD

The MGB axis refers to the bidirectional communication and interaction between the enteric and central nervous systems, which connect the brain and the gut. The MGB axis involves the nervous, endocrine, and immune systems and regulates communication between the brain and the gut by modulating nerve and endocrine cell secretion of hormones, neurotransmitters, mucins, and other signaling molecules, exciting or inhibiting the effector cells in the gastrointestinal tract to maintain overall balance (Jiang et al., 2017; L et al., 2023). The MGB axis is crucial in AD pathophysiological processes, and its mechanisms have been well elucidated (Fig. 1). Altered microbial diversity and derived metabolites in AD disrupt the gut microbiota dynamic balance, leading to microbial–gut epithelial and gut–immune system crosstalk and gut-immune barrier developmental and functional defects (Gasaly, de Vos & Hermoso, 2021). The gut microbiota-derived metabolites worsen through the destruction of the gut mucosal immune system, increased gut permeability (“leaky gut”), and altered bacterial translocation (Jiang et al., 2017), exacerbating systemic inflammatory responses (imbalance of Th17/Treg cells, interleukin [IL]-6 and IL-1β, interferon-gamma [IFN-γ], TNF-α, and TGF-β), which contribute to the adverse effects related to AD pathogenesis (Gasaly, de Vos & Hermoso, 2021; Hur et al., 2020; Yao & Yan, 2020).

Figure 1 The potential role of the microbiota-gut-brain axis in the pathogenesis.

It has been suggested that gut microbiota alterations affect the enteric nervous system (ENS), leading to central nervous system involvement through the increased gut–blood–brain barrier permeability, immune system response activation, and increased gut secretion and derived metabolites, participating in AD pathophysiology (Giau et al., 2018). Gut microbiota changes modulate the peripheral and central nervous systems, altering brain function and providing further evidence for the existence of the MGB axis. Additionally, immune cells secreted by the gut microbiota (microglia and astrocytes) regulate neuroinflammation and are specifically involved in AD development (Rothhammer et al., 2018; Chandra, Sisodia & Vassar, 2023). The blood–brain barrier disruption, increased local and systemic inflammation due to proinflammatory cytokine synthesis, and gut microbiota-secreted signaling molecules are transmitted in the central nervous system through lymphatics and systemic circulation, and these pathological changes are involved in AD pathogenesis (Rieder et al., 2017; van der Eijk et al., 2019). Microbial cell components, such as LPS (MGB axis components), disrupt blood–brain barrier function, activate Toll-like receptor 4, cause epithelial and gut wall inflammation, and lead to gut permeability. This process is essential to the MGB axis in AD (Lin et al., 2020; Khan et al., 2020; Shabbir et al., 2021; Yin et al., 2023). In addition to these pathways, signal pathways, such as the mitogen-activated protein kinase pathway, are also associated with the MGB axis-based mechanisms of AD (Guo et al., 2020b). These bidirectional communication pathways and interactions between metabolites form a complex network system, complicating research on the mechanisms by which the gut microbiota regulates AD (Fig. 1).

The Influence of the Gut Microbiota on Other Nervous Systems

Several studies confirmed the influence of the gut microbiota on other nervous systems, such as in Parkinson’s and Huntington’s diseases (Zhou et al., 2023; Wronka et al., 2023). Gut microbiota changes are crucial in the neuroimmune system, contributing to neurological diseases. Specifically, gut microbiota imbalances and increased gut permeability lead to an overactive immune system, triggering the activation of intestinal neurons and intestinal glial cells and initiating α-synuclein misfolding through the gut-brain axis, accelerating Parkinson’s disease progression (Mulak & Bonaz, 2015). Additionally, experimental studies reported increased gut permeability, a relative abundance of Bacteroidetes, and decreased Firmicutes in the intestinal microbiota of a Huntington’s disease R6/2 mouse model. Meanwhile, altered gut permeability and microbiota composition regulate the interaction between the immune response and the central nervous system, inducing disease (Stan et al., 2020; Sharma et al., 2023). Thus, the gut microbiota is crucial in neurological diseases, participating in their pathophysiology through various mechanisms.

Interaction Between Gut Microbiota and AD Treatment Strategies (Western Medicine, Traditional Chinese Medicine, and External Therapies in Traditional Chinese Medicine)

Interaction between gut microbiota and Western medicine treatment for AD

Currently, drug intervention remains the main approach for treating AD. Medications significantly affect the gut microbiota composition and function, which should not be overlooked. Drugs directly or indirectly influence different gut functions, thus directly or indirectly regulating bacterial metabolism and the ability to modulate bacterial activity and efficacy. The drugs used to treat AD, such as acetylcholinesterase inhibitors and NMDA receptor antagonists (memantine), affect the gut microbiota composition and microbial diversity. It has been suggested that memantine promotes the capture and bactericidal activity of neutrophils induced by pathogenic E. coli in the host gut (Peng et al., 2020; Xiao et al., 2022). Most studies indicated that AD drugs mainly promote gut microbiota and immune-inflammatory functions of derived metabolites, regulating inflammatory responses, stimulating the vagus nerve, and binding to various receptors through the brain-gut axis, thus improving cognitive function.

Interaction between gut microbiota and traditional Chinese medicine treatment for AD

Traditional Chinese medicine is increasingly applied in clinical practice. Traditional Chinese medicine increases gut microbiota species diversity and community diversity and regulates the gut microbiota structure (Table 2). Schisandra chinensis improved gut microbiota dysbiosis in AD rats, promoted SCFA production, regulated the inflammatory metabolic pathways, promoted the growth of beneficial bacteria, and decreased the relative abundance of pathogenic bacteria (Zhao et al., 2016; Wang et al., 2021b; Guo et al., 2020a; Fu, 2023). Furthermore, the relative abundance of OTU345, OTU380 (Lachnospiraceae), and OTU264 (Bacteroidales_S24-7_group) increased significantly in AD patients treated with a compound herbal medicine based on Xiaoyao powder, while the abundance of functional pathways, such as tryptophan metabolism, arginine and proline metabolism, and niacin metabolism, decreased (Weijie, Xingfei & He, 2021). Lachnospiraceae has a strong ability to produce anti-inflammatory butyrate; therefore, its increased abundance plays an important synergistic role in AD treatment (Minter et al., 2017). Additionally, the water extract of Gastrodia elata exerted a good reverse recovery effect on gut microbiota dysbiosis caused by D-galactose and aluminum chloride and increased the abundance of beneficial bacteria, such as Lactobacillus johnsonii, L. murinus, and L. reuteri (Wenbin, 2019). These studies demonstrated that traditional Chinese medicine modulates gut microbiota function and composition, benefiting AD treatment.

Table 2 The relevant herbs and traditional medicines.

TCM formula	Impact on gut microbiota	Impact on cognitive function	Reference	
Huanglian Jiedu Decoction	Prevotellaceae and its genus Prevotellaceae_UCG-001↑, Prevotellaceae_Ga6A1_group, and Parasutterella↑	Suppressed Aβ accumulation, harnessed neuroinflammation, and reversed cognitive impairment	Gu et al. (2021)	
Liuwei Dihuang Bolus	 Bacteroides and Parabacteroides ↓ Acidiphilium↑	Delayed ageing processes, improved cognitive impairments, and balanced the neuroendocrine immunomodulation system	Wang et al. (2019b)	
Xiao yao san	Alpha diversity and the relative abundance of Lachnospiraceae↑	Improved cognitive function, including shortened escaped latency and increased number platforms crossed	Hao et al. (2021)	
Morinda officinalis How	g_fnorank_f_Clostridiales_vadinBB60 group, g_Peptococcus, and g_Ruminococcaceae_UCG_009 ↓	Alleviated memory impairment, reduced toxic Aβ1–42 deposition and neuroinflammation	Xie et al. (2020)	
Icariin (ICA) combined with Panax notoginseng saponins (PNS)	Lactobacillus, Bifidobacterium, and Adlercreutzia ↓	Improved the symptoms of AD patients	Zhang et al. (2020)	
Shu Dihuang	Bacteroidetes, Bacteroidia ↑ Firmicutes and Bacill ↓	Improved cognitive dysfunction and brain pathological changes	Su et al. (2023)	
Chaihu Shugan San	The relative abundance of L. reuteri significantly ↑ the relative abundance of S. xylosus↓	improved the memory deficits and the learning function and ameliorated neuronal injury, synaptic injury, and Aβ deposition in the brain of SAMP8 mice	Li et al. (2023)	

Interaction between gut microbiota and external therapies of traditional Chinese medicine (acupuncture) for AD

Traditional Chinese medicine offers diverse disease treatments. In addition to herbal interventions, numerous external therapies can improve AD clinical manifestations. Research has indicated that acupuncture can balance gut microbiota quantity and composition to suppress peripheral and central nervous system inflammation reactions (Jiang et al., 2021; He et al., 2021). Experimental studies have demonstrated that electroacupuncture increased the abundance of delta-Proteobacteria and epsilon-Proteobacteria in the intestine and improved learning and memory abilities in AD mice (Jiang et al., 2021).

New evidence from animal (Jiang et al., 2021; Yu et al., 2020) and human (Wang et al., 2020) studies supports acupuncture as a candidate method for treating AD. A randomized controlled clinical trial demonstrated that patients in the acupuncture group exhibited increased gut microbiota diversity compared to the non-acupuncture group, and their AD symptoms were improved through brain-gut axis regulation (Kong et al., 2023; Bao et al., 2023). Increased gut microbial diversity and beneficial bacteria and their derived metabolites stimulate the vagus nerve in the gut and enhance the immune response, thus improving brain function. Despite clinical support and ongoing research, the available data and studies in this area and sample sizes are limited. The increased gut microbiota diversity and abundance due to acupuncture have been confirmed, but the specific microbial abundances or the implications of their increase have not been clearly elucidated. Therefore, increased research on the effects of acupuncture and other traditional Chinese medicine external therapies on the gut microbiota is necessary. Thus, external therapies of traditional Chinese medicine, such as acupuncture, are potential alternative approaches for treating AD (Table 3).

Table 3 Interaction between gut flora and acupuncture for Alzheimer’s disease.

Author	Subject of study	n1/n2 (E/C)	Experimental group intervention	Control group intervention and doses	Period of treatment	Outcome measure	Microbiota change	Other outcome	
Jiang et al. (2021)	SAMP8 and SAMR1	8/8	Baihui and Yintang were chose for electroacupuncture	donepezil hydrochloride tablets /Baihui and Yintang were chose for electroacupuncture	15 min/day	Morris water maze/Dentate gyrus of hippocampus/pro-inflammation	the ratio of Bacteroidia and Clostridia ↑	↑ learning and memory abilities ↓IL-1β;IL-6 and TNF-α	
Kong et al. (2023)	Participants with mild AD	53/53	receive acupuncture at Baihui, Sishencong, Shenting, bilateral Neiguan, Shenmen, Zusanli, Taixi, and Sanyinjiao	non-penetrating sham acupuncture	135min/ week 14 weeks	ADAS-cog12/ MMSE/ADCS-ADL, NPI/Gut microbiota will be measured using 16S rRNA tag sequencing.	Experiments in progress	Experiments in progress	
Liao et al. (2021)	APP/PS1 double transgenic mice/male C57BL/6 mice	7/7	receive acupuncture at Baihui Dachangshu and Zusanli	receive acupuncture at Baihui Dachangshu and Zusanli for	15 min/day 5 days per week for 5 weeks.	The Morris water maze/NLRP3/TNF-α/IL-1β/IL-18/the microflora in the feces	↓the relative abundance of Bacteroidetes ↑Firmicutes Patescibacteria and Tenericuteswere ↓the relative abun- dances of Parasutterella and Rikenella ↑ the relative abundances of Candidatus Saccharimonas and Muribaculum	↓ NLRP3/TNF- α/IL-1β/IL-18 ↑ cognitive functions	
Chen et al. (2022)	SD rats/VD model	10/10	receive acupuncture at Baihui Shenshu Dazhui and Zusanli	receive acupuncture at Baihui Shenshu and Dazhui +Probiotics received gavage of Probiotics (2 ml/d containing 2.0 × 109 CFU of live bifidobacterium	30 min/day 4 weeks	The Morris water maze/IL-1β/IL-18/the microflora in the feces	↑Elusimicrobia Firmicutes Proteobacteria Verrucomicrobia ↓the relative abundance of Bacteroidetes	↓IL-1β/IL-18 ↑ cognitive functions	
Xu et al. (2022)	SAMP8 and SAMR1	6/6	Acupuncture points prescribed included Fenglong, Taibai, Taixi, Feiyang and Baihui	Donepezil hydrochloride tablets (5 mg per day)	20 min/day	Aβ1-42 and APP protein expression/MWM/SCFA/the microflora in the feces	↑LPS ↓Firmicutes ↓Streptococcus	↑the learning and memory dysfunction ↓Aβ1-42 and APP protein expression	
Notes.

ADAS-cog12 Alzheimer’s disease Assessment Scale

MMSE Mini-Mental State Examination

ADCS-ADL the Alzheimer’s disease Cooperative Study-Activities

ADAS-cog12 Alzheimer’s disease Assessment Scale

MMSE Mini-Mental State Examination

ADCS-ADL the Alzheimer’s disease Cooperative Study-Activities of Daily Living

NPI Neuropsychiatric Inventory; Gut microbiota will be measured using 16S rRNA tag sequencing

SAMP8 senescence-accelerated mouse prone 8

SAMR1 senescence-accelerated mouse resistant1

LPS lipopolysaccharide

IL-6 Interleukin-6

IL-1β Interleukin-1β

TNF-α Tumor necrosis factor-alpha

SCFAs short chain fatty acids

MWM Morris water maze test

↑ increase

↓ decrease

Clinical Prospects of Gut Microbiota in Therapeutic Applications

The gut microbiota exhibits high dynamism and diversity and the ability for external modulation. The gut microbiota of AD patients exhibits noticeable compositional differences. Controlling diet and introducing specific or mixed active microorganisms regulate the microbiota composition and function to maintain a healthy gut balance. Furthermore, the gut microbiota mediates neuroendocrine activities through metabolic signaling pathways. This mediation highlights the potential of the gut microbiota as a novel target in AD treatment. The understanding of how the gut microbiota influences disease progression and the mechanisms of relevant pharmaceutical effects has progressed significantly. However, the specific role of the microbiota in AD pathogenesis remains unclear. Increasing evidence suggested that microbiota-targeted therapies, such as dietary interventions, FMT, probiotics, prebiotics, and synbiotics, have significant AD-alleviating effects (Fig. 2).

Figure 2 Microbiota-targeting therapies for Alzheimer’s disease.

Dietary interventions

Diet directly influences gut microbiota aggregation and genomic composition, which is a critical factor (Liu et al., 2020). Extensive research has summarized the connections between dietary patterns, gut microbiota, and AD, identifying them as effective candidates for preventing and impeding AD mechanisms and progression. The Mediterranean diet (MD) is characterized by high consumption of fruits, vegetables, legumes, and grains, as well as anti-inflammatory properties, and has long been considered a healthy dietary pattern (Liu et al., 2020; Long-Smith et al., 2020). A systematic review reported that adherence to the MD can reduce the risk of developing AD and help maintain beneficial gut bacteria, demonstrating its preventive role in AD (Solch et al., 2022). A recent meta-analysis of prospective data focusing on older adults reported significant improvements in AD symptoms, specifically in overall cognition and episodic memory, under MD interventions (Loughrey et al., 2017). The MD is associated with gut microbiota diversity and gut inflammation. A positive correlation was indicated between the MD and increased numbers of beneficial microbial species, such as certain Bacteroides species and their anti-inflammatory SCFA metabolites. This alleviation of gut inflammation contributed to reduced neuroinflammation and improved neurocognitive function (McGrattan et al., 2019). Taylor et al. (2017) observed the relationship between carbohydrate intake and AD, demonstrating a positive correlation between carbohydrate intake and increased cerebral amyloid-beta protein load in 26% of participants. Multiple studies suggested dietary recommendations for individuals with AD, including increasing folate, vitamin B6, and vitamin B12 intake while reducing saturated fat, excess dietary fat, and SFAs (Stefaniak et al., 2022; Zhu et al., 2023). Thus, dietary therapy can be a potential adjunctive treatment for AD. However, significant individual variation means that personalized diets should be tailored to maximize therapeutic efficacy.

FMT

FMT is a clinically valuable approach that involves transferring gut microbiota from a healthy donor to restore a dysbiotic gut microbiota in patients, aiming to normalize its composition (Wang et al., 2019a; Grabrucker et al., 2023). Various clinical trials have reported that FMT improved clinical manifestations in AD patients, including cognitive impairments. AD-like phenotypes, such as Iba-1 activation, amyloid-beta protein deposition, and spatial memory decline, were observed in 8-month-old endogenous melatonin reduced (EMR) mice, while FMT increased gut permeability, suppressed systemic inflammation, and alleviated AD-related phenotypes in the mice (Zhang et al., 2022a). Altered gut microbiota abundance is at the core of FMT. FMT increased the Bacteroides species and restored Bacteroidetes abundance in AD patients (Bäuerl et al., 2018; He et al., 2018). Additionally, a human case study involving an AD patient receiving FMT from a healthy 47-year-old woman reported significant improvements in cognitive function as measured by mini-mental state examination (MMSE) and clinical dementia assessment scores (Park et al., 2021). Despite demonstrating efficacy and safety in mice, challenges remain in the clinical application of FMT, including acceptance between recipients and donors, individual immune response mechanism variations, and the influence of lifestyle factors, such as diet, on gut microbiota composition. Future research should address these issues (Table 4).

Table 4 Intestinal bacteria interact with FMT.

Author	Sample size	Donor	Recipient	Transplantation technique	Results	
(Park et al., 2021)
(Li et al., 2023)	Case study (n = 1)	27-year-old healthy man	90-year-old woman with AD and severe CDC	Colonoscopy (60 g of stool suspension for 2 times).	↑ Cognititve function tests (MMSE, MCA and CDR tests) ↑ Microbiota α diversity = Microbiota β diversity ↑ SCFAs	
(Kim et al., 2021) (Zhao et al., 2016)	Mice (n = 8)	5xFAD mice	C57BL/6 mice	Oral gavage (200 µl for 5 consecutive days)	↓ Adult hippocampal neurogenesis and BDNF expression ↑ p21 expression ↑ Microgia activation ↑ TNF-α and IL-1β	
(Sun et al., 2019) (Wang et al., 2021b)	Mice (n = 8)	WT mice	APPswe/PS1dE9 transgenic (Tg) mouse model	Intragastrically (0.2 mL of fresh fecal solution once daily for 4 weeks)	↑ Cognititve function (MWM and ORT tests) ↓ Amyloid β brain deposition (Aβ40 and Aβ42) ↓ Tau protein phosphorylation ↑ Synaptic plasticity (increased PSD-95 and synapsin C) ↓ COX2 and CD11b	
(Elangovan, Borody & Holsinger, 2022) (Guo et al., 2020b)	Mice (n = 16)	B6SJL mice	5xFAD mice	Oral gavage (200 µL of supernatant over a period of seven days)	↑ SCFA and microbiota composition ↑ Cognitive function (increase clearance of cortical Aβ, decreased amyloid burden and amyloid plaque) ↑ spatial memory levels (D1 and FAY) ↑ Mood	
(Park et al., 2022) Fu (2023)	Mice (n = 10)	no gastrointestinal problems or other health problems	(age range, 63–90 years; female, 80%) with dementia and severe CDI	Colonoscopy (60 g of stool suspension).	↑ Cognititve function (MMSE) ↑ clinical symptoms (CDR-SB) ↑ the bacterial richness oogut microbiota (proteobacteria and Bacteroides)	
(Wang et al., 2022) (Weijie, Xingfei & He, 2021)	Mice (n = 16)	Six AD (age range, 72.2 years)/APP/PS1 mice	C57BL/6J mice(Male, 6-week-old)	Infusion (200 µL of bacterial suspension, Once every 2 days for 2 weeks.)	↓ Cognititve function ↑ plasma TMAO ↓ RA of Bacteroidetes ↑ RA of Firmicutes ↑ cortical perk ↑ EIF2α phosphorylation	
(Holsinger & Elangovan, 2020) (Minter et al., 2017)	Mice (n = 16)	healthy, wild type mice	5 XF AD transgenic mice	FMT infusion (for seven days)	↑ Cognititve function ↓ amyloid pathology	
(Zhou et al., 2019) (Wenbin, 2019)	Mice (n = 6)	XAN	AXF and ASF group mice	intragastrical administrated with fecal suspension (1 g/kg body weight)	↑ spatial memory ability (Morris Water Maze Test) ↓ hippocampal neuronal loss ↓ the ratio of Firmicutes/Bacteroidetes the relative abundances of Clostridium IV, Enterohabdus, Coriobacterium, Corynebacterium, Desulfovibrio, and Defluvitalea ↑ the ratio of Methanomassilicoccus, Azoarcus, Phycisphaera, Acetobacteroides, and Alloprevotella	
Notes.

APPSWE/PS1L166P APPPS1-21

BDNF brain derived neurotrophic factor

CDI Clostridioides difficile infection

CDR Clinical Dementia Rating assessment

APPPS1 raised transgenic APPPS1 mice

COX2 cyclooxygenase 2

FMT fecal microbiota transplantation

LPS lipopolysaccharide

MCA Montreal Cognitive Assessment

MMSE Mini-Mental State Examination

MWM Morris water maze test

DI Discrimination Index

CDR-SB Clinical Dementia Rating scale Sum of Boxes

FAY forced alternation Y-maze

ORT object recognition test

MoCA Montreal Cognitive Assessment

SCFAs short chain fatty acids

↑ increase

↓ decrease

Probiotics

Probiotics are supplements of live microorganisms that can promote a balanced gut microbiota in the host, resulting in positive effects (Fuller, 1989; Faulin & Estadella, 2023). Several probiotic strains exerted positive therapeutic effects in AD, with a focus on the genera Bifidobacterium and Lactobacillus. In mouse models, specific strains, such as Bifidobacterium longum A1 and L. helveticus Shirota, effectively counteracted AD (Kobayashi et al., 2017). Furthermore, a randomized controlled trial (RCT) demonstrated that increasing intake of L. acidophilus, L. helveticus, Bifidobacterium, and fermented milk products altered gut microbiota diversity and significantly improved MMSE scores (Leblhuber et al., 2018). Probiotics suppress inflammatory responses. Experimental studies demonstrated that probiotics are important in regulating the immune responses of Th1, Th2, Th17, Treg, natural killer (NK), and B cells (Dargahi et al., 2019). Additionally, some studies demonstrated that probiotics reduced mucosal inflammation by modulating cytokine networks and macrophage tissue and regulating local immune responses (Sichetti et al., 2018).

The oxidative stress response is an important mechanism by which probiotics participate in AD pathological and physiological reactions. Recent research demonstrated that the SLAB51 formulation significantly reduced oxidative stress levels in transgenic AD mice by inducing SIRT-1-dependent mechanisms. In that experiment, researchers used a SIRT-related gene to modulate the response of AD mice to SLAB51 and SLAB50 (Bonfili et al., 2018). Furthermore, integrating L. acidophilus, L. fermentum, L. plantarum, and Bacillus subtilis successfully improved the oxidative stress performance in rats injected with Aβ1-42 (Athari Nik Azm et al., 2018). However, other studies indicated that probiotic intake did not significantly improve AD symptoms. An RCT reported that the intake of two different probiotic capsule formulations did not significantly affect cognitive function or related inflammatory factors in AD patients (Leblhuber et al., 2019). Thus, while some evidence suggests the potential benefits of probiotic intake in improving cognitive function, much data is produced in laboratory settings, and confirmation is needed to establish their clinical usefulness and robustness.

Prebiotics

Prebiotics are sources of nutrition for gut microbiota that can promote beneficial bacterial activity, exerting positive effects on host organs (Ansari et al., 2020). Clinical research supports prebiotic intervention in AD. Fructooligosaccharides (FOS) and mannanoligosaccharides (MOS) are the most common prebiotics used to treat AD, along with compounds from yacon, inulin, and compounds derived from vegetables, herbs, and plants. Additionally, the recently discovered xylooligosaccharides (XOS), which are similar to oligosaccharides, have a wide range of sources and are the most abundant biopolymers in the plant kingdom. XOS exerted significant effects in altering gut microbiota composition and improving AD symptoms (Peterson, 2020; Liu et al., 2021; Arora, Green & Prakash, 2020; Ávila et al., 2020).

Multiple experimental studies indicated that prebiotics achieve their therapeutic effects in AD through various mechanisms. For example, Liu et al. (2021) reported that an 8-week intervention with MOS in 5xFAD AD mice improved cognitive function and spatial memory and reduced anxiety and compulsive behavior. In-depth analysis of the corresponding mechanisms revealed that MOS prevented intestinal barrier integrity damage and LPS leakage, rebuilt the gut microbiota composition, and significantly reduced inflammation by enhancing SCFA formation, achieving the intervention goals (Liu et al., 2021). Furthermore, oral administration of oligosaccharides derived from Marinda officinalis reduced Aβ accumulation in the brains of AD mice, reconstructed oxidative-reductive homeostasis, and reduced overall inflammation, improving memory and learning capabilities (Xu et al., 2020; Chen et al., 2017). Thus, the diverse prebiotics and the mechanisms they use render them a promising therapeutic approach worthy of further development and research.

Synbiotics

Synbiotics combine prebiotics and probiotics, providing substrates for beneficial bacterial proliferation. The synergistic effect of both components contributed to the development of synbiotics. A synbiotic supplement consisting of triptolide-rich plant prebiotic Trifolium pratense leaf extract (TFLA) and probiotics demonstrated stronger survival capacity in transgenic humanized Drosophila melanogaster with BACE1-APP-induced AD phenotypes compared to TFLA or probiotic intervention alone (Westfall, Lomis & Prakash, 2019). Another synbiotic formulation composed of Acetobacter, Delbrueckii, L. fermentum, Leuconostoc, Bifidobacterium, Sporolactobacillus, L. helveticus, Kluyveromyces, and Kluyveromyces marxianus in 4% kefir grain synthesis improved memory, visual-spatial, executive, and linguistic abilities in test subjects and decreased the expression of proinflammatory cytokines (IL-8, IL-12, and TNF-α) while increasing the overall anti/proinflammatory cytokine ratio (Arora, Green & Prakash, 2020; Ton et al., 2020). In summary, synbiotics are a powerful biological tool for treating AD, and the proper selection of probiotic strains and prebiotics lays a solid foundation for formulating synbiotic combinations that confer physiological benefits to the host.

Limitations

This review has several limitations. First, the article length constraints resulted in some aspects not being comprehensively and extensively summarized and analyzed. However, this does not undermine the scholarly nature of this review. Second, this review focused excessively on the relationship between the gut microbiota and AD pathogenesis and treatment while neglecting the links between other reactions in which the gut microbiota is involved, such as intestinal mucosal immune responses and AD. Therefore, this review focused less on mechanisms such as the gut-brain axis and cellular responses. Lastly, there was minimal incorporation and summary of statistical data. Relevant systematic reviews are needed to comprehensively assess and analyze the influence of the gut microbiota on AD and intervention strategies involving the gut microbial community and its derived metabolites in AD to provide a more complete understanding.

Summary and Prospects

The gut microbiota is emerging as a key regulatory factor in human health. Much recent evidence demonstrated that the gut microbial community and its derived metabolites are important in AD pathogenesis. However, this field continues to face significant challenges and difficulties in its future development. This review proposes feasible suggestions to contribute to future advancements in this field:

(1) Further exploration of the causal relationship between the gut microbiota and AD to reveal their underlying connections. Prospective and human case studies are needed to investigate the causality between dysbiosis, AD, and their mechanisms, including drug–microbiota interactions. This review provides a reference for clinicians, including the importance of gut microbiota in AD pathogenesis and the effect of drug–probiotic interactions on improving AD symptoms;

(2) Investigation of the physiological and pathological correlations between the gut-brain axis, gut microbial community, and AD. While preclinical and clinical evidence enabled initial understanding of their mechanisms, whether they exert their effects through other pathways or systems remains unknown;

(3) Consideration of multiple confounding factors to optimize microbiota-based diagnostic and treatment strategies. The composition, proportion, and abundance of gut microbial communities vary due to the uncontrollability of external factors and host individuality. Adverse reactions and complications that might occur after treatment should also be considered. Therefore, the mixed factors behind the benefits and harms of host health should be considered when treating AD. This consideration will lay the foundation for targeted manipulation of AD based on gut microbial communities and their derived metabolites;

(4) Gene sequencing and establishment of a systematic database. Large-scale clinical data research is needed to summarize the genes associated with each symptom and the corresponding common pathogenic and beneficial strains and to isolate these strains through gene sequencing-based microbial cultivation techniques. This research will establish a systematic AD-specific microbial species treatment database.

Unresolved Issues

Although current research and the literature demonstrate the intimate connection between AD and gut microbiota ecosystem stability, many unresolved questions remain in the literature:

(1) How does microbial diversity change at different AD development stages?

(2) How do the brain-gut mechanisms and intestinal mucosal immune responses participate in AD physiological and pathological processes?

(3) Which external factors have the greatest effect on AD progression?

Additionally, intestinal mucosal resistance to external microbiota is individual-specific, so the development of microbiota-targeted therapies should consider the differences between recipients and donors, drug administration timing, and the short- and long-term risks after administration in the complex intestinal environment. Traditional Chinese medicine interventions have yielded promising results in AD diagnosis and treatment, but whether traditional Chinese medicine interventions and microbiota-based therapies differ requires further investigation. The relationship between the microbiota and the host is intricate; therefore, future developments should focus on the relationship. The field continues to face significant challenges, and it is hoped that more research will be conducted and new technologies developed to address these challenges and provide clinicians with more standardized, safe, and effective treatment options.

Additional Information and Declarations

Competing Interests

Author Contributions

Data Availability

The authors declare there are no competing interests.

Xinfu Zou conceived and designed the experiments, performed the experiments, analyzed the data, prepared figures and/or tables, authored or reviewed drafts of the article, and approved the final draft.

Guoqiang Zou conceived and designed the experiments, performed the experiments, analyzed the data, authored or reviewed drafts of the article, and approved the final draft.

Xinyan Zou conceived and designed the experiments, performed the experiments, analyzed the data, prepared figures and/or tables, authored or reviewed drafts of the article, and approved the final draft.

Kangfeng Wang conceived and designed the experiments, performed the experiments, analyzed the data, prepared figures and/or tables, authored or reviewed drafts of the article, and approved the final draft.

Zetao Chen conceived and designed the experiments, performed the experiments, analyzed the data, prepared figures and/or tables, authored or reviewed drafts of the article, and approved the final draft.

The following information was supplied regarding data availability:

This is a literature review.

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
