# Peer review of "Gut microbiota and its metabolites in Alzheimer’s disease: from pathogenesis to treatment"

_PeerJ, doi:10.7717/peerj.17061_

## Round 0.1 · original submission · Major Revisions

Dear Authors,

Thank you for submitting your work to our journal. We have received feedback from peer reviewers, and I would like to inform you that the manuscript is currently categorized under "Moderate" for revisions.

I kindly request you to carefully review the feedback provided by the reviewers and respond to each point raised in detail. In particular, Reviewer 1 emphasized the importance of including specific target genes and citing recent reports. Reviewer 2 suggested the inclusion of information on microbiome shifts due to natural molecules.

It is essential that you address these specific points in your response, providing comprehensive explanations and incorporating the necessary revisions into your manuscript. Additionally, all reviewers have collectively suggested a language check for the manuscript. Therefore, please ensure that your revised work meets the highest standards in terms of language and clarity.

We appreciate your attention to these comments and look forward to receiving your point-by-point responses and the revised version of your manuscript. If you have any questions or require further clarification, please do not hesitate to reach out.

Thank you for your dedication to the improvement of your work. We anticipate reading your revised manuscript soon.

Best wishes.
Dr. Nagendran Tharmalingam
Academic Editor.

Reviewer 1 ·

Basic reporting

no comment

Experimental design

This review paper delves into existing studies that explore the relationship between dysbiosis of gut microbiota and Alzheimer's disease, the role of gut microbiota-derived metabolites in AD patients, and available treatment strategies in both Western and Chinese medicine. Below are my suggestions and comments:

1. The authors assert that the gut microbiota composition differs between AD patients and healthy individuals (Lines 123-124). It would be intriguing to include statistical data or numbers representing various gut microbiota compositions in both diseased and normal individuals.

2. The manuscript could benefit from a discussion on whether specific target genes are affected during microbiota imbalance in AD conditions. Elaborating on this aspect would enhance the comprehensiveness of the paper.

3. Only two papers/studies from the year 2023 are discussed. It would be valuable for the authors to incorporate more recent studies to provide an up-to-date overview of the field.

4. The language throughout the manuscript could be improved for clarity and coherence. Additionally, there is a repetition of the same word in line 133 (decreasing trend... Actinobacteria and Actinobacteria…”) and consider bolding the subheadings to enhance visual organization. This should be addressed during the revision process.

5. This review primarily focuses on Alzheimer's Disease (AD). However, I suggest that the authors consider expanding the scope to include the link between microbiota and other neurological disorders, such as Huntington's disease, Parkinson's disease, and so on (one paragraph).

Validity of the findings

no comment

·

Basic reporting

no comment

Experimental design

no comments

Validity of the findings

no comments

Reviewer 3 ·

Basic reporting

Gut Microbiota and Its Metabolites in Alzheimer's Disease: From Pathogenesis to Treatment
Line no – 133- Term Actinobacteria repeated
Line 143-144- Some more evidence can be added to the point - Variations in the gut microbiota can trigger the onset of Alzheimer's disease (AD), while the pathological changes in AD lead to ecosystem disruption by altering the gut environment
Interaction between Gut Microbiota and Traditional Chinese Traditional herbal Medicine for the treatment of Alzheimer's Disease can be added to the article
The overall article is good; however, the above points can be considered for the betterment of the manuscript.
English grammar should be checked.

Experimental design

no comment

Validity of the findings

no comment

---

## Round 0.2 · accepted · Accept

Dear Authors,

We would like to inform you that after careful consideration by our peers, we are happy to inform you that your work has been accepted for publication with PeerJ. Please work closely with the production team for the proofread copy.

Best wishes.

Dr. Nagendran Tharmalingam
Academic Editor.

Reviewer 1 ·

Basic reporting

no comment

Experimental design

no comment

Validity of the findings

no comment

·

Basic reporting

Authors have carefully addressed all the comments and revised the manuscript accordingly.

Experimental design

No comments

Validity of the findings

No comments

Additional comments

No comments